# Interpretable Diffusion via Information Decomposition

**Xianghao Kong[1] \* , Ollie Liu[2] \*, Han Li[1], Dani Yogatama[2], Greg Ver Steeg[1]**
[1]University of California Riverside, [2]University of Southern California
{xkong016,hli358,gregoryv}@ucr.edu, {zliu2898, yogatama}@usc.edu

## Abstract

Denoising diffusion models enable conditional generation and density modeling of complex relationships like images and text. However, the nature of the learned relationships is opaque making it difficult to understand precisely what relationships between words and parts of an image are captured, or to predict the effect of an intervention. We illuminate the fine-grained relationships learned by diffusion models by noticing a precise relationship between diffusion and information decomposition. Exact expressions for mutual information and conditional mutual information can be written in terms of the denoising model. Furthermore, *pointwise* estimates can be easily estimated as well, allowing us to ask questions about the relationships between specific images and captions. Decomposing information even further to understand which variables in a high-dimensional space carry information is a long-standing problem. For diffusion models, we show that a natural non-negative decomposition of mutual information emerges, allowing us to quantify informative relationships between words and pixels in an image. We exploit these new relations to measure the compositional understanding of diffusion models, to do unsupervised localization of objects in images, and to measure effects when selectively editing images through prompt interventions.

## 1 Introduction

Denoising diffusion models are the state-of-the-art for modeling relationships between complex data like images and text. While diffusion models exhibit impressive generative abilities, we have little insight into precisely what relationships are learned (or neglected). Often, models have limited value without the ability to dissect their contents. For instance, in biology specifying which variables have an effect on health outcomes is critical. As AI advances, more principled ways to probe learned relationships are needed to reveal and correct gaps between human and AI perspectives.

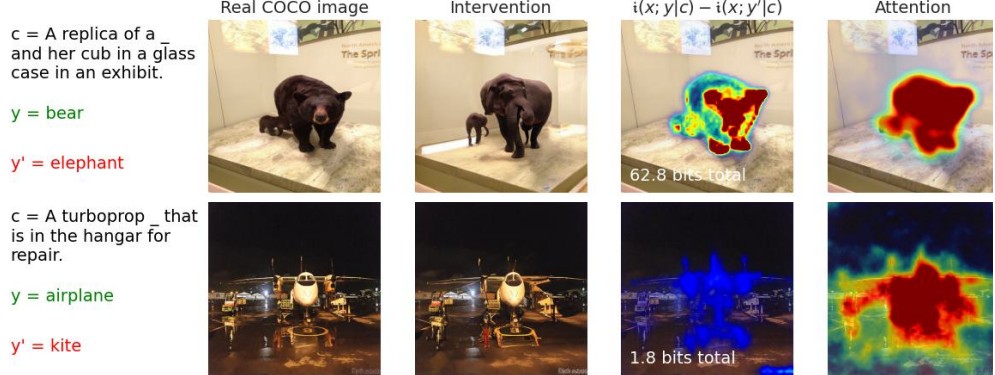

Figure 1: We start (left) with a real image from the COCO dataset. We do a "prompt intervention" (§3.3) to generate a new image. Next we show conditional mutual information, illustrated using our pixel-wise decomposition, and attention maps for the modified word. Top row shows an image where prompt intervention has an effect, while in the bottom row it has little effect. Conditional mutual information reflects the effect of intervention while attention does not.

Quantifying the relationships learned in a complex space like text and images is difficult. Information theory offers a black-box method to gauge how much information flows from inputs to outputs. This work proceeds from the novel observation that diffusion models naturally admit a simple and versatile information decomposition that allows us to pinpoint information flows in fine detail which allows us to understand and exploit these models in new ways.

For denoising diffusion models, recent work has explored how *attention* can highlight how models depends on different words during generation (Tang et al., 2022; Liu et al., 2023; Zhao et al., 2023; Tian et al., 2023; Wang et al., 2023; Zhang et al., 2023; Ma et al., 2023; He et al., 2023). Our information-theoretic approach diverges from attention-based methods in three significant ways. First, attention requires not just white-box access but also dictates a particular network design. Our approach abstracts away from architecture details, and may be useful in the increasingly common scenario where we interact with large generative models only through black-box API access (Ramesh et al., 2022). Second, while attention is engineered toward specific tasks such as segmentation Tang et al. (2022) and image-text matching He et al. (2023), our information estimators can adapt to diverse applications. As an illustrative example, in §3.1 we automate the evaluation of compositional understanding for Stable Diffusion Rombach et al. (2022). Third, information flow as a dependence measure better captures the effects of interventions. Attention within a neural network does not necessarily imply that the final output depends on the attended input. Our Conditional Mutual Information (CMI) estimator correctly reflects that a word with small CMI will not affect the output (Fig. 1). We summarize our main contributions below.

- We show that denoising diffusion models directly provide a natural and tractable way to decompose information in a fine-grained way, distinguishing relevant information at a per-sample (image) and per-variable (pixel) level. The utility of information decomposition is validated on a variety of tasks below.
- We provide a better quantification of compositional understanding capabilities of diffusion models. We find that on the ARO benchmark Yuksekgonul et al. (2022), diffusion models are significantly underestimated due to sub-optimal alignment scores.
- We examine how attention and information in diffusion models localize specific text in images. While neither exactly align with the goal of object segmentation, information measures more effectively localize abstract words like adjectives, adverbs, and verbs.
- How does a prompt intervention modify a generated image? It is often possible to surgically modify real images using prompt intervention techniques, but sometimes these interventions are completely ignored. We show that CMI is more effective at capturing the effects of intervention, due to the ability to take contextual information into account.

## 2 Methods: Diffusion is Information Decomposition

### 2.1 Information-theoretic Perspective on Diffusion Models

A diffusion model can be seen as a noisy channel that takes samples from the data distribution, $\boldsymbol{x} \sim p(X = \boldsymbol{x})$, and progressively adds Gaussian noise, $\boldsymbol{x}_\alpha \equiv \sqrt{\sigma(\alpha)}\boldsymbol{x} + \sqrt{\sigma(-\alpha)}\boldsymbol{\epsilon}$, with $\boldsymbol{\epsilon} \sim \mathcal{N}(0, \mathbb{I})$ (a variance preserving Gaussian channel with log SNR, $\alpha$, using the standard sigmoid function). By learning to reverse or *denoise* this noisy channel, we can generate samples from the original distribution (Sohl-Dickstein et al., 2015), a result with remarkable applications (Ramesh et al., 2022). The Gaussian noise channel has been studied in information theory since its inception (Shannon, 1948). A decade before diffusion models appeared in machine learning, Guo et al. (2005) demonstrated that the information in this Gaussian noise channel, $I(X; X_\alpha)$, is *exactly* related to the mean square error for optimal signal reconstruction. This result was influential because it demonstrated for the first time that *information-theoretic* quantities could be related to *estimation of optimal denoisers*. In this paper, we are interested in extending this result to other mutual information estimators, and to *pointwise* estimates of mutual information. Our focus is not on learning the reverse or denoising process for generating samples, but instead to *measure relationships* using information theory.

For our results, we require the optimal denoiser, or Minimum Mean Square Error (MMSE) denoiser for predicting $\boldsymbol{\epsilon}$ at each noise level, $\alpha$.

$$\hat{\boldsymbol{\epsilon}}_\alpha(\boldsymbol{x}) \equiv \arg\min_{\bar{\boldsymbol{\epsilon}}(\cdot)} \mathbb{E}_{p(\boldsymbol{x}), p(\boldsymbol{\epsilon})} \left[ \|\boldsymbol{\epsilon} - \bar{\boldsymbol{\epsilon}}(\boldsymbol{x}_\alpha)\|^2 \right] \tag{1}$$

Note that we predict the noise, $\boldsymbol{\epsilon}$, but could equivalently predict $\boldsymbol{x}$. This optimal denoiser is exactly what diffusion models are trained to estimate. Instead of using denoisers for generation, we will see how to use them for measuring information-theoretic relationships.

For the denoiser in Eq. 1, the following expression holds exactly.

$$-\log p(\boldsymbol{x}) = \tfrac{1}{2} \int \mathbb{E}_{p(\boldsymbol{\epsilon})} \left[ \|\boldsymbol{\epsilon} - \hat{\boldsymbol{\epsilon}}_\alpha(\boldsymbol{x}_\alpha)\|^2 \right] d\alpha + const \tag{2}$$

The value of the constant, $const$, will be irrelevant as we proceed to build Mutual Information (MI) estimators and a decomposition from this expression. This expression shows that solving a denoising *regression* problem (which is easy for neural networks) is *equivalent* to density modeling. No differential equations need to be referenced or solved to make this exact connection, unlike the approaches appearing in Song et al. (2020) and McAllester (2023). The derivation of this result in Kong et al. (2022) closely mirrors Guo et al. (2005)'s original result, and is shown in App. A for completeness. This expression is extremely powerful and versatile for deriving fine-grained information estimators, as we now show.

## 2.2 MUTUAL INFORMATION AND POINTWISE ESTIMATORS

Note that Eq. 2 also holds with arbitrary conditioning. Let $\boldsymbol{x}, \boldsymbol{y} \sim p(X = \boldsymbol{x}, Y = \boldsymbol{y})$ and $\hat{\boldsymbol{\epsilon}}_\alpha(\boldsymbol{x}_\alpha | \boldsymbol{y})$ be the optimal denoiser for $p(\boldsymbol{x} | \boldsymbol{y})$ as in Eq. 1. Then we can write the conditional density as follows.

$$-\log p(\boldsymbol{x} | \boldsymbol{y}) = \tfrac{1}{2} \int \mathbb{E}_{p(\boldsymbol{\epsilon})} \left[ \|\boldsymbol{\epsilon} - \hat{\boldsymbol{\epsilon}}_\alpha(\boldsymbol{x}_\alpha | \boldsymbol{y})\|^2 \right] d\alpha + const \tag{3}$$

This directly leads to an estimate of the following useful Log Likelihood Ratio (LLR).

$$\log p(\boldsymbol{x} | \boldsymbol{y}) - \log p(\boldsymbol{x}) = \tfrac{1}{2} \int \mathbb{E}_{p(\boldsymbol{\epsilon})} \left[ \|\boldsymbol{\epsilon} - \hat{\boldsymbol{\epsilon}}_\alpha(\boldsymbol{x}_\alpha)\|^2 - \|\boldsymbol{\epsilon} - \hat{\boldsymbol{\epsilon}}_\alpha(\boldsymbol{x}_\alpha | \boldsymbol{y})\|^2 \right] d\alpha \tag{4}$$

The LLR is the integrated reduction in MMSE from conditioning on auxiliary variable, $\boldsymbol{y}$. The mutual information, $I(X; Y)$ can be defined via this LLR, $I(X; Y) \equiv \mathbb{E}_{p(\boldsymbol{x}, \boldsymbol{y})} \left[ \log p(\boldsymbol{x} | \boldsymbol{y}) - \log p(\boldsymbol{x}) \right]$. We write MI using information theory notation, where capital $X, Y$ are used to refer to functionals of random variables with the distributions $p(\boldsymbol{x}, \boldsymbol{y})$ Cover & Thomas (2006). MI is an average measure of dependence, but we are often interested in the strength of a relationship for a single point, or *pointwise information*. Pointwise information for a specific $\boldsymbol{x}, \boldsymbol{y}$ is sometimes written with lowercase as $\mathrm{i}(\boldsymbol{x}; \boldsymbol{y})$ and is defined so that the average recovers MI (Finn & Lizier, 2018). Fano (1961) referred to what we call "mutual information" as "average mutual information", and considered what we call pointwise mutual information to be the more fundamental quantity. Pointwise information has been especially influential in NLP Levy & Goldberg (2014).

$$I(X; Y) = \mathbb{E}_{p(\boldsymbol{x}, \boldsymbol{y})} [\mathrm{i}(\boldsymbol{x}; \boldsymbol{y})] \qquad \textit{Defining property of pointwise information}$$

Pointwise information is not unique and both quantities below satisfy this property.

$$\mathrm{i}^s(\boldsymbol{x}; \boldsymbol{y}) \equiv \tfrac{1}{2} \int \mathbb{E}_{p(\boldsymbol{\epsilon})} \left[ \|\boldsymbol{\epsilon} - \hat{\boldsymbol{\epsilon}}_\alpha(\boldsymbol{x}_\alpha)\|^2 - \|\boldsymbol{\epsilon} - \hat{\boldsymbol{\epsilon}}_\alpha(\boldsymbol{x}_\alpha | \boldsymbol{y})\|^2 \right] d\alpha$$

$$\mathrm{i}^o(\boldsymbol{x}; \boldsymbol{y}) \equiv \tfrac{1}{2} \int \mathbb{E}_{p(\boldsymbol{\epsilon})} \left[ \|\hat{\boldsymbol{\epsilon}}_\alpha(\boldsymbol{x}_\alpha) - \hat{\boldsymbol{\epsilon}}_\alpha(\boldsymbol{x}_\alpha | \boldsymbol{y})\|^2 \right] d\alpha \tag{5}$$

The first **standard definition** comes from using $\log p(\boldsymbol{x} | \boldsymbol{y}) - \log p(\boldsymbol{x})$ written via Eq. 4. The second more compact definition is derived using the **orthogonality principle** in §B. $\mathrm{i}^s$ has higher variance due to the presence of extra $\boldsymbol{\epsilon}$ terms, while $\mathrm{i}^o$ has lower variance and is always non-negative. We will explore both estimators, but find the lower variance version that exploits the orthogonality principle is generally more useful (see §C.2). Note that while (average) MI is always non-negative, pointwise MI can be negative as $\mathrm{i}^s(\boldsymbol{x}; \boldsymbol{y}) = \log p(\boldsymbol{x} | \boldsymbol{y}) - \log p(\boldsymbol{x}) < 0$ occurs when the observation of $\boldsymbol{y}$ makes $\boldsymbol{x}$ appear *less* likely. We can say negative pointwise information signals a "misinformative" observation (Finn & Lizier, 2018).

All these expressions can be given conditional variants, where we condition on a random variable, $C$, defining the context. CMI and its pointwise expression can be related as $I(X; Y | C) = \mathbb{E}_{p(\boldsymbol{x}, \boldsymbol{y}, \boldsymbol{c})} [\mathrm{i}(\boldsymbol{x}; \boldsymbol{y} | \boldsymbol{c})]$. The pointwise versions of Eq. 5 can be obtained by conditioning all the denoisers on $C$, e.g., $\hat{\boldsymbol{\epsilon}}_\alpha(\boldsymbol{x}_\alpha | \boldsymbol{y}) \to \hat{\boldsymbol{\epsilon}}_\alpha(\boldsymbol{x}_\alpha | \boldsymbol{y}, \boldsymbol{c})$.

## 2.3 PIXEL-WISE INFORMATION DECOMPOSITION

The pointwise information, $i(\boldsymbol{x}; \boldsymbol{y})$, does not tell us which variables, $x_j$'s, are informative about which variables, $y_k$. If $\boldsymbol{x}$ is an image and $\boldsymbol{y}$ represents a text prompt, then this would tell us which parts of the image a particular word is informative about. One reason that information decomposition is highly nontrivial is that scenarios can arise where information in variables is synergistic, for example (Williams & Beer, 2010). Our decomposition instead proceeds from the observation that the correspondence between information and MMSE leads to a natural decomposition of information into a sum of terms for each variable. If $\boldsymbol{x} \in \mathbb{R}^n$, we can write $i(\boldsymbol{x}; \boldsymbol{y}) = \sum_{j=1}^{n} i_j(\boldsymbol{x}; \boldsymbol{y})$ with:

$$
i_j^s(\boldsymbol{x}; \boldsymbol{y}) \equiv \frac{1}{2} \int \mathbb{E}_{p(\boldsymbol{\epsilon})} \left[ (\boldsymbol{\epsilon} - \hat{\boldsymbol{\epsilon}}_\alpha(\boldsymbol{x}_\alpha))_j^2 - (\boldsymbol{\epsilon} - \hat{\boldsymbol{\epsilon}}_\alpha(\boldsymbol{x}_\alpha|\boldsymbol{y}))_j^2 \right] d\alpha
$$
$$
i_j^o(\boldsymbol{x}; \boldsymbol{y}) \equiv \frac{1}{2} \int \mathbb{E}_{p(\boldsymbol{\epsilon})} \left[ (\hat{\boldsymbol{\epsilon}}_\alpha(\boldsymbol{x}_\alpha) - \hat{\boldsymbol{\epsilon}}_\alpha(\boldsymbol{x}_\alpha|\boldsymbol{y}))_j^2 \right] d\alpha
$$

(6)

In other words, both variations of pointwise information can be written in terms of squared errors, and we can decompose the squared error into the error for each variable. For pixel-wise information for images with multiple channels, we sum over the contribution from each channel.

We can easily extend this for conditional information. Let $\boldsymbol{x}$ represent a particular image, $\boldsymbol{y} = \{y_*, \boldsymbol{c}\} = \{\text{"object"}, \text{"a person holding an \_"}\}$. Then we can estimate $i_j(\boldsymbol{x}; y_*|\boldsymbol{c})$, which represents the information that word $y_*$ has about variable $\boldsymbol{x}_j$, conditioned on the context, $\boldsymbol{c}$. To get estimates using Eq. 6, we just add conditioning on $\boldsymbol{c}$ on both sides. Denoising images conditioned on arbitrary text is a standard task for diffusion models. An example of the estimator is shown in Fig. 1, where the highlighted region represents the pixel-wise value of $i_j^o(\boldsymbol{x}; y_*|\boldsymbol{c})$ for pixel $j$.

## 2.4 NUMERICAL INFORMATION ESTIMATES

All information estimators we have introduced require evaluating a one-dimensional integral over an infinite range of SNRs. To estimate in practice we use importance sampling to evaluate the integral as in (Kingma et al., 2021; Kong et al., 2022). We use a truncated logistic for the importance sampling distribution for $\alpha$. Empirically, we find that contributions to the integral for both very small and very large values of $\alpha$ are close to zero, so that truncating the distribution has little effect. Unlike MINE Belghazi et al. (2018) or variational estimators (Poole et al., 2019) the estimators presented here do not depend on optimizing a direct upper or lower bound on MI. Instead, the estimator depends on finding the MMSE of both the conditional and unconditional denoising problems, and then combining them to estimate MI using Eq. 4. However, these two MMSE terms appear with opposite signs. In general, any neural network trained to minimize MSE may not achieve the global minimum for either or both terms, so we cannot guarantee that the estimate is either an upper or lower bound. In practice, neural networks excel at regression problems, so we expect to achieve reasonable estimates. In all our results we use pretrained diffusion models which have been trained to minimize mean square under Gaussian noise, as required by Eq. 1 (different papers differ in how much weight each $\alpha$ term receives in the objective, but in principle the MMSE for each $\alpha$ is independent, so the weighting shouldn't strongly affect an expressive enough neural network).

## 3 RESULTS

Section 2 establishes a precise connection between optimal denoisers and information. For experiments we consider diffusion models as approximating optimal denoisers which can be used to estimate information. All our experiments are performed with pre-trained latent space diffusion models Rombach et al. (2022), Stable Diffusion v2.1 from Hugging Face unless otherwise noted. Latent diffusion models use a pre-trained autoencoder to embed images in a lower resolution space before doing typical diffusion model training. We always consider $\boldsymbol{x}$ as the image in this lower dimensional space, but in displayed images we show the images after decoding, and we use bilinear interpolation when visualizing heat maps at the higher resolution. See §D for experiment details and links to open source code.

## 3.1 Relation Testing with Pointwise Information

First, we consider information decomposition at the "pointwise" or per-image level. We make use of our estimator to compute summary statistics of an image-text pair and quantify qualitative differences across samples. As a novel application scenario, we apply our pointwise estimator to analyze compositional understanding of Stable Diffusion on the ARO benchmark Yuksekgonul et al. (2022). Referring readers to Yuksekgonul et al. (2022) for more detailed descriptions, the ARO benchmark is a suite of discriminative tasks wherein a VLM is commissioned to align an image $\boldsymbol{x}$ with its ground-truth caption $\boldsymbol{y}$ from a set of perturbations $\mathcal{P} = \{\tilde{\boldsymbol{y}}_j\}_{j=1}^M$ induced by randomizing constituent orders from $\boldsymbol{y}$. The compositional understanding capability of a VLM is thus measured by the accuracy:

$$\mathbb{E}_{\boldsymbol{x},\boldsymbol{y}}[\mathbb{1}\big(\boldsymbol{y} = \arg\max_{\boldsymbol{y}' \in \{\boldsymbol{y}\} \bigcup \mathcal{P}} s(\boldsymbol{x}, \boldsymbol{y}')\big)],$$

where $s : \mathcal{X} \times \mathcal{Y} \to \mathbb{R}$ is an alignment score. For contrastive VLMs, $s$ is chosen to be the cosine similarity between encoded image-text representations. We choose $s(\boldsymbol{x}, \boldsymbol{y}) \equiv \mathrm{i}^o(\boldsymbol{x}; \boldsymbol{y})$ as our score function for diffusion models. In contrast, He et al. (2023) compute $s$ as an aggregate of latent attention maps, while Krojer et al. (2023) adopt the negative MMSE. In Table 1, we report performances of OpenCLIP Ilharco et al. (2021) and Stable Diffusion 2.1 Rombach et al. (2022) on the ARO benchmark, while controllilng model checkpoints with the same text encoder for fair comparison.

Table 1: Accuracy (%) of Stable Diffusion and its OpenCLIP backbone on the ARO Benchmark. ⋆: conducts additional fine-tuning with compositional-aware hard negatives.

| Method | VG-A | VG-R | COCO | Flickr30k |
|---|---|---|---|---|
| OpenCLIP Ilharco et al. (2021) | 64.6 | 51.4 | 32.8 | 40.5 |
| DiffITM Krojer et al. (2023) | 62.9 | 50.0 | 23.5 | 33.2 |
| HardNeg-DiffITM⋆ Krojer et al. (2023) | 67.6 | 52.3 | 34.4 | 48.6 |
| Info. (Ours) | **72.0** | **69.1** | **40.1** | **49.3** |

We observe that Stable Diffusion markedly improves in compositional understanding over Open-CLIP. Since the text encoder is frozen, we can attribute this improvement solely to the denoising objective and the visual component. More importantly, our information estimator significantly outperforms MMSE Krojer et al. (2023), decidedly proving that previous works underestimate compositional understanding capabilities of diffusion models. Our observation provides favorable evidence for adapting diffusion models for discriminative image-text matching (He et al., 2023). However, these improvements are smaller compared to contrastive pre-training approaches that make use of composition-aware negative samples (Yuksekgonul et al., 2022).

In Table 2 we report a subset of fine-grained performances across relation types, highlighting those with over 30% performance improvement in green and those with over 30% performance decrease in magenta. Interestingly, our highlighted categories correlate well with performance changes incurred by composition-aware negative training, despite using different CLIP backbones (c.f. Table 2 of Yuksekgonul et al. (2022)). Most improvements are attributed to verbs that associate subjects with conceptually distinctive objects (e.g. "*A boy sitting on a chair*"). Our observation suggests that improve-

Table 2: Selected results on VG-R.

| | Info. (↑) | OpenCLIP (↑) |
|---|---|---|
| **Accuracy (%)** | 69.1 | 51.4 |
| beneath | 50.0 | 90.0 |
| covered in | 14.3 | 50.0 |
| feeding | 50.0 | 100.0 |
| grazing on | 60.0 | 30.0 |
| sitting on | 78.9 | 49.7 |
| wearing | 84.1 | 44.9 |

ments in compositional understanding may stem predominantly from these "low-hanging fruits", and partially corroborate the hypothesis in Rassin et al. (2023), which posits that incorrect associations between entities and their visual attributes are attributed to the text encoder's inability to encode linguistic structure. We refer readers to §D.1 for complete fine-grained results, implementation details, and error analysis.

## 3.2 Pixel-wise Information and Word Localization

Next, we explore "pixel-wise information" that words in a caption contain about specific pixels in an image. According to §2.2 and §2.3, it naturally leads us to consider two potential approaches

for validating this nuanced relationship. The first one entails concentrating solely on the mutual information between the object word and individual pixels. The second approach involves investigating this mutual information when given the remaining context in the caption, i.e., conditional mutual information. As the success of our experiment relies heavily on the alignment between images and text, we carefully filtered two datasets, COCO-IT and COCO-WL from the MSCOCO (Lin et al., 2015) validation dataset. For specific details about dataset construction, please refer to §E. Meanwhile, we also provide an image-level information analysis on these two datasets in §C.1, and visualize the diffusion process and its relation to information for 10 cases in §C.2.

### 3.2.1 VISUALIZE MUTUAL INFORMATION AND CONDITIONAL MUTUAL INFORMATION

Given an image-text pair $(\boldsymbol{x}, \boldsymbol{y})$ where $\boldsymbol{y} = \{y_*, \boldsymbol{c}\}$, we compute pixel-level mutual information as $\mathfrak{i}^o_j(\boldsymbol{x}; y_*)$, and the conditional one as $\mathfrak{i}^o_j(\boldsymbol{x}; y_*|\boldsymbol{c})$ for pixel $j$, from Eq. 6. Eight cases are displayed in Fig. 2, and we put more examples in §C.3 and §C.4.

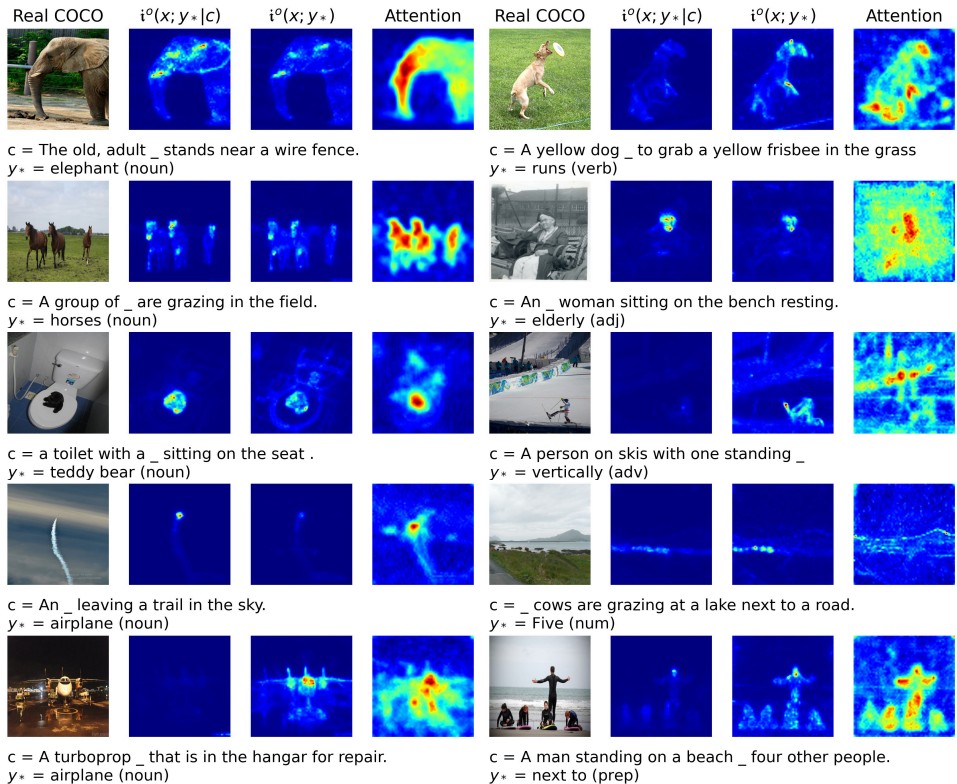

Figure 2: Examples of localizing different types of words in images. The left half presents noun words, while the right half displays abstract words.

In Fig. 2, the visualizations of pixel-level MI and CMI are presented using a common color scale. Upon comparison, it becomes evident that CMI has a greater capacity to accentuate objects while simultaneously diminishing the background. This is attributed to the fact that MI solely computes the relationship between noun word $y_*$ and pixels, whereas CMI factors out context-related information from the complete prompt-to-pixel information. However, there are occasional instances of the opposite effect, as observed in the example where '$y_*$ = airplane' (left bottom in Fig. 2). In this case, it appears that CMI fails to highlight pixels related to 'airplane', while MI succeeds. This discrepancy arises from the presence of the word 'turboprop' in the context. Therefore the context, $\boldsymbol{c}$, accurately describes the content in the image and 'airplane' adds no additional information. Compared to attention, CMI and MI qualitatively appear to focus more on fine details of objects, like the eyes, ears, and tusks of an elephant, or the face of horses. For object words, we can use a segmentation task to make more quantitative statements in §3.2.2.

We also explore whether pixel-level CMI or MI can provide intriguing insights for other types of entities. Fig. 2 (right) presents four different types of entities besides nouns, including verbs, adjectives, adverbs, numbers and prepositions. These words are quite abstract, and even through manual annotation, it can be challenging to precisely determine the corresponding pixels for them. The visualization results indicate that MI gives intuitively plausible results for these abstract items, especially adjectives and adverbs, that highlight relevant finer details of objects more effectively than attention. Interestingly, MI's ability to locate abstract words within images aligns with the findings presented in §3.1 regarding relation testing (Table 10).

### 3.2.2 LOCALIZING WORD INFORMATION IN IMAGES

The visualizations provided above offer an intuitive demonstration of how pixel-wise CMI relates parts of an image to parts of a caption. Therefore, our curiosity naturally extends to whether this approach can be applied to word localization within images. Currently, the prevalent evaluation involves employing attention layers within Stable Diffusion models for object segmentation Tang et al. (2022); Tian et al. (2023); Wang et al. (2023). These methodologies heavily rely on attention layers and meticulously crafted heuristic heatmap generation. It's worth highlighting that during image generation, the utilization of multi-scale cross-attention layers allows for the rapid computation of the Diffuse Attention Attribution Map (DAAM) Tang et al. (2022). This not only facilitates segmentation but also introduces various intriguing opportunities for word localization analyses. We opted for DAAM as our baseline choice due to its versatile applicability, and the detailed experimental design is documented in the §D.2. We use mean Intersection over Union (mIoU) as the evaluation

Table 3: Unsupervised Object Segmentation mIoU (%) Results on COCO-IT

| Method | 50 steps | 100 steps | 200 steps |
| --- | --- | --- | --- |
| Whole Image Mask | 14.94 | 14.94 | 14.94 |
| Attention Tang et al. (2022) | 34.52 | 34.90 | 35.35 |
| Conditional Mutual Info. | 32.31 | 33.24 | 33.63 |
| Attention+Information | **42.46** | **42.71** | **42.84** |

metric for assessing the performance of pixel-level object segmentation. Table 3 illustrates that, in contrast to attention-based methods, pixel-wise CMI proves less effective for object segmentation, with an error analysis appearing in Table 9. The attention mechanism in DAAM combines high and low resolution image features across the multi-scale attention layers, akin to Feature Pyramid Networks (FPN) Lin et al. (2017), facilitating superior feature fusion. CMI tends to focus more on the specific details that are unique to the target object rather than capturing the overall context of it. Although pixel-level CMI did not perform exceptionally well in object segmentation, the results from 'Attention+Information' clearly demonstrate that the information-theoretic diffusion process Kong et al. (2022) does enhance the capacity of attention layers to capture features.

**Discussion: attention versus information versus segmentation**   Looking at the heatmaps, it is clear that some parts of an object can be more informative than others, like faces or edges. Conversely, contextual parts of an image that are not part of an object can still be informative about the object. For instance, we may identify an airplane by its vapor trail (Fig. 2). Hence, we see that neither attention nor information perfectly aligns with the goal of segmentation. One difference between attention and mutual information is that when a pixel pays "attention" to a word in the prompt, it *does not* imply that modifications to the word would modify the prompt. This is best highlighted with conditional mutual information, where we see that an informative word ("jet") may contribute little information in a larger context ("turboprop jet"). To highlight this difference, we propose an experiment where we test whether intervening on words changes a generated image. Our hypothesis is that if a word has low CMI, then its omission should not change the result. For attention, on the other hand, this is not necessarily true.

### 3.3 SELECTIVE IMAGE EDITING VIA PROMPT INTERVENTION

Diffusion models have gained widespread adoption by providing non-technical users with a natural language interface to create diverse, realistic images. Our focus so far has been on how well diffusion

models understand structure in *real* images. We can connect real and generated images by studying how well we can *modify* a real image, which is a popular use case for diffusion models discussed in §4. We can validate our ability to measure informative relationships by seeing how well the measures align with effects under prompt intervention.

For this experiment, we adopt the perspective that diffusion models are equivalent to continuous, invertible normalizing flows (Song et al., 2020). In this case, the denoising model is interpreted as a score function, and an ODE depending on this score function smoothly maps the data distribution to a Gaussian, or vice versa. The solver we use for this ODE is the 2nd order deterministic solver with 100 steps from Karras et al. (2022). We start with a real image and prompt, then use the (conditional) score model to map the image to a Gaussian latent space. In principle, this mapping is invertible, so if we reverse the dynamics we will recover the original image. We see in §C.5 that the original image is almost always recovered with high fidelity.

Next, we consider adding an intervention. While doing the reverse dynamics, we *modify* the (conditional) score by changing the denoiser prompt in some way. We focus in experiments on the effects of omitting a word, or swapping a word for a categorically similar word ("bear" → "elephant", for example). Typically, we find that much of the detail in the original image is preserved, with only the parts that relate to the modified word altered. Interestingly, we also find that in some cases interventions to a word are seemingly ignored (see Fig. 1, 3).

We want to explore whether attention or information measures predict the effect of interventions. When intervening by omitting a word, we consider the conditional (pointwise) mutual information $i(\boldsymbol{x}; y|\boldsymbol{c})$ from Eq. 5, and for word swaps we use a difference of CMIs with $y, y'$ representing the swapped word, as in Fig. 1. For attention, we aggregate the attention corresponding to a certain word during generation using the code from Tang et al. (2022). We find that a word can be ignored *even if the attention heatmap highlights an object related to the word*. In other words, attention to a word does not imply that it affects the generated outcome. One reason for this is that a word may provide little information beyond that in the context. For instance, in Fig. 3, a woman in a hospital is assumed to be on a bed, so omitting this word has no effect. CMI, on the other hand, correlates well with the effect of intervention, and "bed" in this example has low CMI.

To quantify our observation that conditional mutual information better correlates with the effect of intervention than attention models, we measure the Pearson correlation between a score (CMI or attention heatmap) and the effect of the intervention using L2 distance between images before and after intervention. To get an image-level correlation, we correlated the aggregate scores per image across examples on COCO100-IT. We also consider the pixel-level correlations between L2 change per pixel and metrics, CMI and attention heatmaps. We average these correlations over all images and report the results in Table 4. CMI is a much better predictor of changes at the image-level, which reflects the fact that it directly quantifies what additional information a word contributes after taking context into account. At the per-pixel level, CMI and attention typically correlated very well when changes were localized, but both performed poorly in cases where a small prompt change led to a global change in the image. Results visualizing this effect are shown along with additional experiments with word swap interventions in §C.5. Small dependence, as measured by information, correctly implied small effects from intervention. Large dependence, however, can lead to complex, global changes due to the nonlinearity of the generative process.

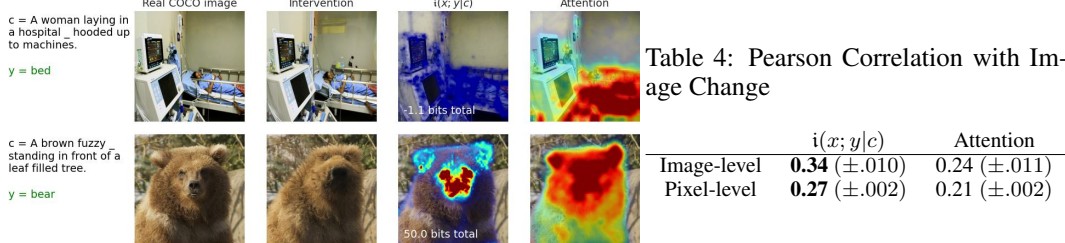

Table 4: Pearson Correlation with Image Change

|  | $i(x; y|c)$ | Attention |
|---|---|---|
| Image-level | **0.34** ($\pm$.010) | 0.24 ($\pm$.011) |
| Pixel-level | **0.27** ($\pm$.002) | 0.21 ($\pm$.002) |

Figure 3: COCO images edited by omitting words from the prompt. Conditional mutual information better reflects the actual changes in the image after intervention. Table shows Pearson correlation between metrics (CMI or attention) versus L2 changes in image after intervention, at the image-level and pixel-level as discussed in §3.3.

## 4    RELATED WORK

**Visual perception via diffusion models**: Diffusion models' success in image generation has piqued interest in their text-image understanding abilities. A variety of pipeline designs are emerging in the realm of visual perception and caption comprehension, offering fresh perspectives and complexities. Notably, ODISE introduced a pipeline that combines diffusion models with discriminative models, achieving excellent results in open-vocabulary segmentation (Xu et al., 2023). Similarly, OVDiff uses diffusion models to sample support images for specific textual categories and subsequently extracts foreground and background features (Karazija et al., 2023). It then employs cosine similarity, often known as the CLIP (Radford et al., 2021) filter, to segment objects effectively. MaskDiff (Le et al., 2023) and DifFSS (Tan et al., 2023) introduce a new breed of conditional diffusion models tailored for few-shot segmentation. DDP, on the other hand, takes the concept of conditional diffusion models and applies it to dense visual prediction tasks (Ji et al., 2023). VDP incorporates diffusion for a range of downstream visual perception tasks such as semantic segmentation and depth estimation (Zhao et al., 2023). Several methods have begun to explore the utilization of attention layers in diffusion models for understanding text-image relationships in object segmentation (Tang et al., 2022; Tian et al., 2023; Wang et al., 2023; Zhang et al., 2023; Ma et al., 2023).

**Image editing via diffusion**: Editing real images is an area of growing interest both academically and commercially with approaches using natural text, similar images, and latent space modifications to achieve desired effects (Nichol et al., 2021; Mao et al., 2023; Liu et al., 2023; Balaji et al., 2022; Kawar et al., 2023; Mokady et al., 2023). Su et al. (2022)'s approach most closely resembles the procedure used in our intervention experiments. Unlike previous work, we focus not on edit quality but on using edits to validate the learned dependencies as measured by our information estimators.

**Interpretable ML**: Traditional methods for attribution based on gradient sensitivity rather than attention (Sundararajan et al., 2017; Lundberg & Lee, 2017) are seldom used in computer vision due to the well-known phenomenon that (adversarial) perturbations based on gradients are uncorrelated with human perception (Szegedy et al., 2014). Information-theoretic approaches for interpretability based on information decomposition (Williams & Beer, 2010) are mostly unexplored because no canonical approach exists (Kolchinsky, 2022) and existing approaches are completely intractable for high-dimensional use cases (Reing et al., 2021), though there are some recent attempts to decompose *redundant information* with neural networks (Kleinman et al., 2021; Liang et al., 2023). Our decomposition decomposes an information contribution from each variable, but does not explicitly separate unique and redundant components. "Interpretability" in machine learning is a fuzzy concept which should be treated with caution (Lipton, 2018). We adopted an *operational interpretation* from information theory, which considers $y \rightarrow x$ as a noisy channel and asks how many bits of information are communicated, using diffusion models to characterize the channel.

## 5    CONCLUSION

The eye-catching image generation capabilities of diffusion models have overshadowed the equally important and underutilized fact that they also represent the state-of-the-art in density modeling (Kingma et al., 2021). Using the tight link between diffusion and information, we were able to introduce a novel and tractable information decomposition. This significantly expands the usefulness of neural information estimators (Belghazi et al., 2018; Poole et al., 2019; Brekelmans et al., 2023) by giving an interpretable measure of fine-grained relationships at the level of individual samples and variables. While we focused on vision tasks for ease of presentation and validation, information decomposition can be particularly valuable in biomedical applications like gene expression where we want to identify informative relationships for further study (Pepke & Ver Steeg, 2017). Another promising application relates to contemporaneous works on *mechanistic interpretability* (Wang et al., 2022; Hanna et al., 2023, *inter alia*) which seeks to identify "circuits"—a subgraph of a neural network responsible for certain behaviors—by ablating individual network components to observe performance differences. For language models, differences are typically measured as the change in total probability of interested vocabularies; whereas metric design remains an open question for diffusion models. Our analyses indicate that the CMI estimators are apt for capturing compositional understanding and localizing image edits. For future work, we are interested in exploring their potential as metric candidates for identifying relevant circuits in diffusion models.

## 6    ETHICS STATEMENT

Recent developments in internet-scale image-text datasets have raised substantial concerns over their lack of privacy, stereotypical representation of people, and political bias (Birhane et al., 2021; Peng et al., 2021; Schuhmann et al., 2021, *inter alia*). Dataset contamination entails considerable safety ramifications. First, models trained on these datasets are susceptible to memorizing similar pitfalls. Second, complex text-to-image generation models pose significant challenges to interpretability and system monitoring (Hendrycks et al., 2023), especially given the increasing popularity in blackbox access to these models (Ramesh et al., 2022). These risks are exacerbated by these models' capability to synthesize photorealistic images, forming an "echo chamber" that reinforces existing biases in dataset collection pipelines. We, as researchers, shoulder the responsibility to analyze, monitor, and prevent the risks of such systems at scale.

We believe the application of our work has meaningful ethical implications. Our work aims to characterize the fine-grained relationship between image and text. Although we primarily conduct our study on entities that do not explicitly entail societal implications, our approach can be conceivably adapted to attribute prompt spans responsible for generating images that amplify demographic stereotypes (Bianchi et al., 2023), among many other potential risks. Aside from detecting known risks and biases, we should aim to design approaches that automatically identify system anomalies. In our image generation experiments, we observe changes in mutual information to be inconsistent during prompt intervention. It is tempting to hypothesize that these differences may be reflective of dataset idiosyncrasies. We hope that our estimator can contribute to a growing body of work that safeguards AI systems in high-stake application scenarios (Barrett et al., 2023), as diffusion models could extrapolate beyond existing text-to-image generation to sensitive domains such as protein design (Watson et al., 2023), molecular structure discovery (Igashov et al., 2022), and interactive decision making (Chi et al., 2023).

Finally, it is imperative to address the ethical concerns associated with the use of low-wage labor for dataset annotation and evaluation (Perrigo, 2023). Notably, existing benchmarks for assessing semantic understanding of image generation resort to manual evaluation (Conwell & Ullman, 2022; Liu et al., 2022; Saharia et al., 2022). In §3.1 we adapt our estimator for compositional understanding, and aim to develop an automated metric for evaluation on a broader spectrum of tasks.

### ACKNOWLEDGMENTS

GV thanks participants of the DEMICS workshop hosted at MPI Dresden for valuable feedback on this project.

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
