# OpenReview forum: "Interpretable Diffusion via Information Decomposition"
_ICLR.cc/2024/Conference — ICLR 2024 poster_

### Official Review · Reviewer_U1vn · 2023-10-27

**Soundness:** 3 good
**Presentation:** 2 fair
**Contribution:** 3 good
**Rating:** 6
**Confidence:** 4

**Summary:**

This paper proposes to investigate the fine-grained relationships learned by diffusion models by noticing a precise relationship between diffusion and information decomposition.  It exploits these new relations to measure the compositional understanding of diffusion models, to do unsupervised localization of objects in images, and to measure effects of prompt interventions.

**Strengths:**

1. It is interesting to view diffusion models from the information theory prespective. For examle, exact expressions for mutual information and conditional mutual information can be expressed in terms of the denoising model.
2. Compared with attention, the proposed mutual information seems more exact in identifying the relationship between the text and image. Massive visualization in the supplement demonstrate the effectiveness of this method.

**Weaknesses:**

1. It is still unclear of the concrete implementation. Without access to the network, but just with API, how to get the mutual information and corresponding visualization.
2. What is the application field. Attention-based methods are widely applied to image editing and show impressive results. What is the advantage of this decomposition method, compared to attetion.
3. The paper is poor structured and hard to read.

**Questions:**

Refer to the weakness and questions above.

---

> ### Author Response · Authors · 2023-11-17
>
> Thank you for your thorough review and thoughtful questions! We answer your questions below. Let us know how to further improve the paper!
>
>
> - **Q1: It is still unclear of the concrete implementation. Without access to the network, but just with API, how to get the mutual information and corresponding visualization.**
>
> Thanks for your question! We provide our demo code detailing our implementation for image-level and pixel-level MI/CMI in the updated supplementary material. Our information maps don't require the details of the denoiser (the U-Net for Stable Diffusion) but just use the output of the denoiser directly, e.g. predicted noise $\hat \epsilon_\alpha$, to calculate MI/CMI as an integral of MMSE under a bunch of logSNR $\alpha$. Thus, our MI/CMI will benefit from a more powerful denoiser in the future or a mixture of experts. However, attention maps require extracting features from self-attention or cross-attention layers and delicately designing a heuristic way to calculate heatmaps, which is hard to generalize.
>
> - **Q2: What is the application field? Attention-based methods are widely applied to image editing and show impressive results. What is the advantage of this decomposition method, compared to attention.**
>
> We believe a key strength of our approach is faithful evaluation. On compositional understanding benchmarks, our estimator is significantly better aligned with ground truth compared to other estimators (c.f. the Table in our Scientific Finding section of general response). An application could be image retrieval based on text, for example, along with interpretable maps about why/which parts of an image matched.
>
> - **Q3: The paper is poorly structured and hard to read.**
>
> Thank you for your concern! We will update our presentation in our updated version. Meanwhile, we hope that our added information theory preliminaries and contributions can address some of your concerns.

---

### Official Review · Reviewer_gEHf · 2023-11-01

**Soundness:** 2 fair
**Presentation:** 2 fair
**Contribution:** 3 good
**Rating:** 6
**Confidence:** 5

**Summary:**

This work discusses the interpretability of denoising diffusion models, which are used for conditional generation and density modeling of complex relationships like images and text. The authors aim to understand the relationships learned by these models by establishing a connection between diffusion and information decomposition. They provide exact expressions for mutual information and conditional mutual information in terms of the denoising model. The paper also introduces methods to quantify informative relationships between words and pixels in an image. The authors utilize these relationships to measure the compositional understanding of diffusion models, perform unsupervised localization of objects in images, and measure effects when editing images through prompt interventions.

**Strengths:**

- The paper provides an approach to illuminate the fine-grained relationships learned by diffusion models.
- The authors introduce a natural non-negative decomposition of mutual information, which can be beneficial for understanding high-dimensional data.
- The paper offers practical applications, such as unsupervised localization of objects in images.
- The research uses information theory to provide a black-box method for understanding relationships in complex spaces like text and images.

**Weaknesses:**

- The method is relatively trivial, replacing word in the prompt and comparing the difference in pixel level
- lack of comprehensive comparison with attention based method

**Questions:**

- The example in first row of Fig. 1 seems to have difference other than the bear and elephant, why does the heat map only show difference in salient objects? Other examples have similar questions, but Fig. 1 is most obvious.
- What's the advantage compared to attention based method?

**Details Of Ethics Concerns:**

None.

---

> ### Author Response · Authors · 2023-11-17
>
> Thank you for your critique! We hope to resolve your concerns with additional clarifications on our method and experimental design.
>
>
> - **W1: The method is relatively trivial, replacing word in the prompt and comparing the difference in pixel level**
>
> We are glad that the reviewer found the method so simple that it appears trivial. Mutual information estimation is an extremely challenging problem, with many mathematically complex approaches proposed in the literature. As far as we are aware, ours is the first proposal to estimate MI using diffusion models. However, another diffusion-based method has appeared simultaneously in [1]. We will include this reference in our final version. But we would like to call the reviewer's attention to their final estimator, Eq. 21, which is not nearly as elegant/trivial to understand as ours and is valid only for (average) mutual information but does not generalize to a pointwise version as ours does.
>
> - **W2: Lack of comprehensive comparison with attention based method**
>
> We included attention baselines on all tasks that involved word/pixel level localization (Sec. 3.2, 3.3). The main goal of our method, as discussed in the top-level response, is measuring the total strength of relationships between image and text. We show SOTA results for this task in Sec. 3.1. Attention has not, to our knowledge, ever been used as a measure of total dependence between images and prompts because the attention distribution is a normalized distribution over tokens which must always sum to 1.
>
> - **Q1: The example in the first row of Fig. 1 seems to have differences other than the bear and elephant, why does the heat map only show differences in salient objects? Other examples have similar questions, but Fig. 1 is most obvious.**
>
> Thanks for the question! A simple reason is just that we threshold the color scale, in addition to your points that these shouldn't exactly correspond.
>
> - **Q2: What's the advantage compared to attention based methods?**
>
> We clarify our contributions in the general response. In summary, we propose a mutual information estimator that can be easily obtained from denoising diffusion models. With Stable Diffusion, we find our estimator can provide faithful evaluation for compositional understanding, quantify effect sizes of image edits, and localizing phrases in images.
>
> [1] Franzese et al. 2023. MINDE: Mutual Information Neural Diffusion Estimation. ArXiv

---

> ### Comment · Reviewer_gEHf · 2023-11-21
> **Thanks for response**
>
> Thank you for detailed response to all my concerns and questions.
>
> After reviewing the rebuttal and other reviews feedback, I decide to increase my score for this paper.
>
> Good work.

---

> > ### Author Response · Authors · 2023-11-22
> >
> > We appreciate your positive and timely followup!

---

### Official Review · Reviewer_S4G8 · 2023-11-01

**Soundness:** 3 good
**Presentation:** 3 good
**Contribution:** 3 good
**Rating:** 6
**Confidence:** 3

**Summary:**

Denoising diffusion models are used for complex data relationships like images and text, and this paper has revealed a link between these models and information decomposition, enabling precise characterization of relationships, including mutual information, which can be used to understand and measure the relationships between specific elements in high-dimensional data like words and image pixels.

**Strengths:**

This paper provides a new perspective to explore the relationship between image and text in the Generation field, and quantify compositional understanding capabilities of diffusion models at a certain level.

**Weaknesses:**

However, the primary drawback lies in the paper's somewhat inadequate presentation, making comprehension challenging. Additionally, a limitation is the insufficient experimentation, which fails to demonstrate how this method could enhance the generation process. To conclude, I strongly advise the authors to undergo a significant revision.

**Questions:**

1. Inadequate preliminary information on related information theory makes Section 2 difficult to comprehend.
2. Clicking on citations does not redirect to the bibliography, making it challenging to locate the correct reference.
3. Apart from the formations, understanding the implementation of this method is challenging, necessitating more detailed explanations from the authors.
4. Improved demonstration of how this method enhances the generation process in experiments would be beneficial.
5. Does Section 6 exceed the limitation of 9 pages?

---

> ### Author Response · Authors · 2023-11-17
>
> Thank you for your critique and detailed list of questions! We hope to resolve your concerns and questions below.
>
> - **W1: However, the primary drawback lies in the paper's somewhat inadequate presentation, making comprehension challenging.**
>
> We appreciate your constructive feedback on the presentation. We have updated Section 2 to include more background on information theory, and will aim to improve presentation throughout.
>
> - **W2: Additionally, a limitation is the insufficient experimentation, which fails to demonstrate how this method could enhance the generation process.**
>
> We apologize for causing confusion! We would like to invite you to read our Paper objectives and Contributions sections in the general response to clarify that enhancing generation is not the goal. In summary, we believe our core contribution is an information-theoretic estimator that can be obtained directly from a denoising diffusion model. While our approach is not geared towards generation, we nevertheless believe this is an important topic: measuring text-image relationships is a challenging task, and existing approaches are often model- or task-specific. Please refer to our general response for additional discussions and results.
>
> - **Q1: Inadequate preliminary information on related information theory makes Section 2 difficult to comprehend.**
>
> We have updated our manuscript with additional background in information theory in Section 2.1 and in Appendix B.
>
> - **Q2: Clicking on citations does not redirect to the bibliography, making it challenging to locate the correct reference.**
>
> Thank you for bringing this to our attention! Indeed, the reference hyperlinks ceased to function when we split the appendix from the main body of our manuscript. We leave this unchanged in our manuscript to prevent excessive triggers of pdfdiff, but will address this issue in subsequent updates.
>
> - **Q3: Apart from the formations, understanding the implementation of this method is challenging, necessitating more detailed explanations from the authors.**
>
> Thank you for your interest in understanding our implementation. We have provided code in our supplementary material, and we will release our code on GitHub after the anonymity period.
>
> - **Q4: Improved demonstration of how this method enhances the generation process in experiments would be beneficial.**
>
> Please see our discussion in the second quote block of this response.
>
> - **Q5: Does Section 6 exceed the limitation of 9 pages?**
>
> According to the ICLR 2024 Author Guide (https://iclr.cc/Conferences/2024/AuthorGuide): “The optional ethic statement will not count toward the page limit, but should not be more than 1 page.”

---

> > ### Comment · Reviewer_S4G8 · 2023-11-21
> >
> > Thanks for the authors' further explanation. After carefully reading the rebuttal and revised paper, all my concerns have been addressed. Especially, sorry for the misunderstanding about the ethics statement. I decided to increase my score, and hope this paper can be published at ICLR.
> >
> > On a side note, in the initial stages, I hesitated to respond to the rebuttal due to a lack of feedback from the reviewers of my own paper, which was a source of frustration. However, I have chosen to adhere to my principles and act as a responsible reviewer, regardless of those guys. Meanwhile, I admire your responsible AC who who has emphasized the importance of the reviewing process.

---

> > > ### Author Response · Authors · 2023-11-21
> > >
> > > We sincerely appreciate your positive and timely feedback! Wish you all the best in receiving updates on your own submission.

---

### Official Review · Reviewer_qKVB · 2023-11-01

**Soundness:** 3 good
**Presentation:** 3 good
**Contribution:** 3 good
**Rating:** 6
**Confidence:** 3

**Summary:**

The paper introduces the application of information theory to interpret text-to-image diffusion models. The authors present a way to measure and utilize the mutual information between the image and the text conditioning to visualize "where" a specific prompt maps to. They also demonstrate how this mapping from text to pixels can be used to relate images with text and localize words as well as help in interpretability by visualizing the effects of the text prompt in the generation process.

**Strengths:**

- The paper introduces a novel approach to measuring the relationship between the text conditioning and the image pixels in text-to-image diffusion models. Previous attempts that focused on interpreting the cross-attention operations within the model were architecture-specific and usually required additional processing or fine-tuning to translate attention into meaningful signals. In contrast, the authors show that their information-theoretic approach can be easily interpreted without additional overhead.

- The experiments section clearly demonstrates both the advantages and disadvantages of the proposed approach. The paper paints a clear picture of when the mutual information can be useful and where attention can be used instead.

- The method is well-grounded, with Section 2 presenting the approach and necessary background thoroughly. The paper is overall well-written.

**Weaknesses:**

- The authors present this paper as an approach to interpretability for text-to-image diffusion models but there is no clear demonstration of where this could be applied other than better visualizations of text-image relations. It would be informative to discuss (or even demonstrate) whether the mutual information can be exploited to better align text-to-image models with human preferences during inference time or fine-tuning. The broader scope of interpreting the generation process by relating pixels to words is missing from the paper and could greatly benefit it.

**Questions:**

- How is the attention extracted in Figure 2? The proposed per-pixel mutual information averages the result over all timesteps of the diffusion process. However, the attention seems to be extracted from a single forward pass of the model. That could be disadvantageous as the attention mechanism highlights different scales of features at different timesteps. In section 3.2.2 the authors acknowledge the existence of a  "feature pyramid" in the attention but is this again for a single time step of the diffusion process?

---

> ### Author Response · Authors · 2023-11-17
>
> Thank you for your insightful comments on interpretability! We agree these are valid concerns, and wish to use this opportunity to discuss them.
>
> - **W1: The authors present this paper as an approach to interpretability for text-to-image diffusion models but there is no clear demonstration of where this could be applied other than better visualizations of text-image relations.**
>
> The main value of our approach is for estimating mutual information for relationship testing, which has many applications and for which we get SOTA results in an image-language similarity matching task in Sec. 3.1. A surprising benefit of this particular mutual information method, in stark contrast to other MI estimators, is that it is possible to interpret the discovered relationships at a word and pixel level.
>
> - **W2: It would be informative to discuss (or even demonstrate) whether the mutual information can be exploited to better align text-to-image models with human preferences during inference time or fine-tuning**
>
> We invite you to review the table in the Scientific Finding section of our general response. Our INFO metric is more faithful in reflecting the compositional understanding capabilities of diffusion models; it reveals that contemporaneous works underestimate DMs in this department.
>
> We may presumably adapt our more aligned metric as an unnormalized reward and/or a procedure that filters text-to-image generation results. These filtered results can serve as AI feedback to subsequently improve DMs in compositional understanding via supervised fine-tuning.
>
> - **W3: The broader scope of interpreting the generation process by relating pixels to words is missing from the paper and could greatly benefit it.**
>
> We appreciate your perspective! While the focus of this work is on evaluating text-image relationships with our INFO metric, we believe INFO may contribute to the mechanistic interpretability of the generation process as well. A large body of work in mechanistic interpretability of language models identifies circuits by attributing a performance metric to certain network components. For LMs, these metrics are often implemented as total probabilities of desired token predictions [1][2]; yet they remain elusive for diffusion models. We hope the INFO metric can serve as a faithful and theoretically-grounded metric for similar works of mechanistic interpretability for diffusion models, a future direction which we identify in the Conclusion section of our manuscript.
>
> - **Q1: How is the attention extracted in Figure 2? The proposed per-pixel mutual information averages the result over all timesteps of the diffusion process. However, the attention seems to be extracted from a single forward pass of the model. That could be disadvantageous as the attention mechanism highlights different scales of features at different timesteps. In section 3.2.2 the authors acknowledge the existence of a "feature pyramid" in the attention but is this again for a single time step of the diffusion process?**
>
> Thank you for the question. We use a real image as input, add noise to it, and then denoise it step by step until we get the reconstructed image. The attention heatmaps are generated by the DAAM method [3], which is a wrapper applied on the UNet that tracks the features of the attention layer in the whole denoising process. DAAM extracts features from multi-scale attention layers (feature pyramid) in the UNet at a series of time steps. It upscales all intermediate attention feature arrays to the original image size using bicubic interpolation, then sums them over the heads, layers, and time steps. For a fair comparison, in our experiments, we set the steps as 100.
>
> [1] Wang et al. 2023. Interpretability in the Wild: a Circuit for Indirect Object Identification in GPT-2 Small. ICLR
>
> [2] Hanna et al. 2023. How does GPT-2 compute greater-than?: Interpreting mathematical abilities in a pre-trained language model. NeurIPS
>
> [3] Tang et al. 2022. What the DAAM: Interpreting Stable Diffusion Using Cross Attention. ACL.

---

> > ### Comment · Reviewer_qKVB · 2023-11-21
> >
> > Thank you for the detailed responses to my comments. Given that my concerns were mostly about demonstrating applicability, which I believe you had already done your best to address in the main text and that my score was already positive, I will be keeping my rating as is.

---

> > > ### Author Response · Authors · 2023-11-22
> > >
> > > Thank you for your positive comments. We will continue to improve our paper and study mechanistic interpretability of diffusion models in future works.

---

### Official Review · Reviewer_ajkd · 2023-11-01

**Soundness:** 3 good
**Presentation:** 1 poor
**Contribution:** 3 good
**Rating:** 6
**Confidence:** 4

**Summary:**

This paper proposes a method to visualize the relationship between image and text caption based on the mutual information (MI) estimated by a pre-trained diffusion model. The core observation is that the log probability density can be estimated up to constant by integrating the diffusion-model loss over timesteps (McAllester, 2023), which enables the MI between image and text-caption to be computed by the difference of the estimated log density from the conditional and the unconditional inference result of a text-to-image diffusion model. While the original MI is the average over the samples, the authors consider per-sample (point-wise) MI and further decompose it to per-pixel MI, which gives rise to a visualization of which region in the image is related with a certain word. In experiments, the proposed visualization method showed superior interpretation over the attention-based method.

**Strengths:**

The paper considers an important topic in text-to-image diffusion models. Analyzing the influence of each word to the generated image has been attracting many researchers, since it serves as an essential tool to come up with ideas for improving or debugging the semantic alignment of the generated image and for developing image-editing methods.

While a large body of literature exploits the cross-attention maps for the problem, this paper considers a new approach acts on pixels. The approach itself is quite similar to the classifier-free guidance that contrasts the conditional and unconditional inference results, which has been used in most of the diffusion models, but have strength by grounding on information theory.

Also, experimental results demonstrate the proposed method gives better interpretability compared to the attention-based methods.

**Weaknesses:**

A weak point of this paper is that the main manuscript seems to be not self-contained. While each of the following can be rather minor, they together drops readability.

- References:
    - The proposed method is heavily relying on the log-density estimator of McCallester, 2023, but lacks details. Further explanation on the practical behavior of this estimator and how this estimator can be derived, at least in abstract sense, would help the understanding.
- Rather important figures and tables are in appendix.
    - There is a fair amount of discussion in the main manuscript with Table 9 and Figure 10, but these are on the appendix thus making difficult to read.
- Dataset information
    - CoCo-IT is the main dataset used for evaluation, but how it has been filtered from CoCo is not described at least briefly.

**Questions:**

- Are the “standard definition” and the “orthogonality principle” the full list of the decomposition possibilities? If not, we might need some justification that why these two are better suited as pointwise information.
    - The main manuscript mentions that both $i^s$ and $i^o$ will be explored. Can you locate where the $i^s$ results are?
- Is there a notable relationship between $i^o$ and the classifier-free guidance, considering their similarity in computation? For example, if the guidance scale gets larger, does it make a enhanced visualization or a larger MI between the text and image?
- The Eq. (5) and (6) contains expectation over the noise, $\epsilon$. But different noises will make dramatically different images. How are the visualizations in Fig. 2 are generated, well aligned with the image?

---

> ### Author Response · Authors · 2023-11-17
>
> Thank you for considering our research an important topic and for your insightful comments contrasting our estimators against attention-based methods! Please see our response below.
>
> - **W1: The proposed method is heavily relying on the log-density estimator of McCallester, 2023, but lacks details. Further explanation on the practical behavior of this estimator and how this estimator can be derived, at least in an abstract sense, would help the understanding.**
>
> Thank you for raising this concern. Our information decomposition builds on ITDiffusion [1] rather than McCallester’s work (clarified in the revision). Unlike the approaches of Song, McAllester, and others, our approach does not require complex differential equation solvers for connecting diffusion and density estimation tasks. We added an appendix on the detailed derivation of the density estimator from [1] in an appendix, to make the paper more self-contained.
>
> - **W2: There is a fair amount of discussion in the main manuscript with Table 9 and Figure 10, but these are on the appendix thus making it difficult to read.**
>
> We appreciate your critique! We have moved part of the results from Table 9 and referred example in Figure 10 to our main body.
>
>
> - **W3: CoCo-IT is the main dataset used for evaluation, but how it has been filtered from CoCo is not described at least briefly.**
>
> We described the details of the preprocessing steps for the COCO-IT, COCO100-IT, and COCO-WL datasets in Appendix E.
>
> - **Q1: Are the “standard definition” and the “orthogonality principle” the full list of the decomposition possibilities? If not, we might need some justification as to why these two are better suited as pointwise information.**
>
> This is a good point, it is possible to define a whole continuum of different pointwise estimators by simply adding any zero-mean random variable to the estimate. The “standard” definition is special by convention, as it is the difference of two density estimates. The “orthogonal” definition is special because it is the minimum variance unbiased estimator.
>
> - **Q1.1: The main manuscript mentions that both $i^s$ and $i^o$ will be explored. Can you locate where the $i^s$ results are?**
>
> Yes, we have explored $i^s$ and $i^o$ at image-level (Fig. 5) as well as pixel-level (Fig. 8) in the Appendix C.2.
>
>
> - **Q2: Is there a notable relationship between i^o and the classifier-free guidance, considering their similarity in computation? For example, if the guidance scale gets larger, does it make a enhanced visualization or a larger MI between the text and image?**
>
> This is a great question. The form of the MI and classifier-free guidance formula both include the difference between a conditional estimator and an unconditional one. In this sense, we can interpret the density model with the modified score function using guidance as being one that artificially boosts the mutual information of the original density function. It would be interesting to explore these types of score-based information manipulations further.
>
> - **Q3: The Eq. (5) and (6) contains expectation over the noise, $\epsilon$, But different noises will make dramatically different images. How are the visualizations in Fig. 2 are generated, well aligned with the image?**
>
> Thanks for your question. The calculation of CMI/MI is just an integral of MMSE differences, and MMSE is an expectation over MSE between gaussian noise and the predicted one, not the noise itself. Additionally, we repeat the denoising on multiple sampled noises.
>
> [1] Kong et al. 2023 Information-Theoretic Diffusion ICLR

---

### Official Review · Reviewer_EMcD · 2023-11-08

**Soundness:** 2 fair
**Presentation:** 2 fair
**Contribution:** 2 fair
**Rating:** 6
**Confidence:** 2

**Summary:**

This work presents a new method for information decomposition at both pointwise and pixel-wise level via well-trained diffusion models, utilizing the results such that there is an exact connection between mutual information and MMSE denoising estimators under Gaussian noise, as well as the exact correspondence between denoising and density modeling. The work demonstrates the ability of such information decomposition to measure the compositional understanding of diffusion models, to perform unsupervised localization of objects in images, and to measure effects of selective image editing via prompt interventions. The proposed metric, *Conditional Mutual Information (CMI) estimators*, is displayed as an alternative to attention in capturing the effect of text interventions in image generation.

**Strengths:**

1. The finding that attention to a word doesn’t necessarily imply this word would affect the generated outcome is interesting, and Figure 1 shows how CMI can be helpful in interpreting the model for the cases where attention cannot.
2. The authors present a standalone section to discuss ethical implications of the models they are working on, as well as how their method could help deal with the potential risks.

**Weaknesses:**

1. The walkthrough of some key results that help construct the proposed information decomposition approach is too brief: it would be nice to have a more in-depth review of these results, for example, the derivation of Eq. (2), the part of diffusion model parameterization that’s related to MMSE, etc. Meanwhile, the experiment section is relatively lengthy: some parts could be moved to the appendix, leaving the contents most closely connected to information decomposition in the main paper.
2. Error bars in Table 3 for the comparison between CMI and attention are not reported.
3. Source code is not shared to facilitate reproducibility.

**Questions:**

1. Could the authors give a quick example of how $i^o(\mathbf{x};\mathbf{y})$ from Eq. (5) is computed? Under the DDPM parameterization, the noise network predicts the forward noise, thus it makes sense to start with a real image $\mathbf{x}$, sample $\mathbf{\epsilon}$ to obtain $\mathbf{x}_{\alpha}$, and compute the L2 term: this utilizes the forward diffusion. Meanwhile, the goal is to decompose the *generated* samples, which uses the reverse diffusion. Therefore, it’s a bit unclear on how these mutual information estimators are calculated.
2. The authors mentioned that Eq. (2) would hold for an “optimal denoiser”: what does it mean for a denoiser to be optimal?

---

> ### Author Response · Authors · 2023-11-17
>
> Thank you for noting a key difference between our INFO estimators and attention-based methods. We’re encouraged by your kind words on our ethics statement! Please see our response below.
>
> - **W1: The walkthrough of some key results that help construct the proposed information decomposition approach is too brief: it would be nice to have a more in-depth review of these results, for example, the derivation of Eq. (2), the part of diffusion model parameterization that’s related to MMSE, etc. Meanwhile, the experiment section is relatively lengthy: some parts could be moved to the appendix, leaving the contents most closely connected to information decomposition in the main paper.**
>
> Thank you for your interest in our theoretical results. We have updated Section 2 to provide additional background on information theory and our derivation.
>
> - **W2:  Error bars in Table 3 for the comparison between CMI and attention are not reported.**
>
> We have updated our Table 3 (now Table 4) in our manuscript with (standard errors) for pixel-level experiments. We observe INFO and attention-based methods to be relatively stable. We will update image-level results in our subsequent updates.
>
>  |           | $i(x; y \| c)$ |  Attention |
>  |:-------------------|:--------:|:--------:|
>  |     Image-level     |   **0.50**   |   0.31   |
>  |      Pixel-level     |   **0.31** ($\pm 0.017$)   |   0.27 ($\pm 0.019$)   |
>
> - **W3: Source code is not shared to facilitate reproducibility.**
>
> We provide an example implementation of our INFO estimators (MI and CMI) in the supplementary for your reference. We will provide open-source implementations after the anonymity period.
>
> - **Q1: Could the authors give a quick example of how $i^o(x;y)$  from Eq. (5) is computed? Under the DDPM parameterization, the noise network predicts the forward noise, thus it makes sense to start with a real image $x$,  sample $\epsilon$  to obtain $x_\alpha$  and compute the L2 term: this utilizes the forward diffusion. Meanwhile, the goal is to decompose the generated samples, which uses the reverse diffusion. Therefore, it’s a bit unclear on how these mutual information estimators are calculated.**
>
> You say that "the goal is to decompose the generated samples, which uses the reverse diffusion", but we want to strongly emphasize that this is **not** the goal. We want to decompose the information in **any image, real or generated**. Therefore, our measure always uses the "forward diffusion". That is, we always start with an image, add noise, and then measure the error of the denoiser to recover the signal. This doesn't require access to a reverse diffusion process that generates the image.
>
> - **Q2: The authors mentioned that Eq. (2) would hold for an “optimal denoiser”: what does it mean for a denoiser to be optimal?**
>
> The 'optimal denoiser' means the denoiser minimizes the MMSE (Eq.1) for recovering $x$ in this noisy channel. We clarified this in the revised text.
>
> [1] Kong et al. 2023 Information-Theoretic Diffusion ICLR

---

> > ### Comment · Reviewer_EMcD · 2023-11-21
> >
> > Thank you for addressing my questions and comments. I decide to keep my current score.

---

> > > ### Author Response · Authors · 2023-11-22
> > >
> > > Thank you for your positive feedback. We appreciate your comments and suggestions on improving the clarity of our approach :-)

---

### Author Response · Authors · 2023-11-17

We thank the reviewers for their insightful reviews. We are glad the reviewers found our problem interesting (EMcD, qKVB, U1vn, ajkd), our approach theoretically grounded and scientifically sound (qKVB, gEHf, ajkd), and our experiments extensive (U1vn, ajkd).

Some reviewers have requested clarification on our mutual information estimators (INFO). In short, the focus of our approach is measuring relationships using a novel connection between diffusion models and mutual information. Our unsupervised approach leads to SOTA results on a vision-language relation testing benchmark, beating out even contemporaneous papers that were fine-tuned for this task. Below, we clarify our objective and contributions, adding background information, improving presentation and reproducibility.

---

### Author Response · Authors · 2023-11-17
**General Response (1/2)**

# Paper Objectives
Our work proposes INFO as a principled and denoiser-agnostic estimator to measure complex relationships between image and text. We first derive INFO as a principled estimator grounded in information theory, naturally yielding two quantitative metrics, mutual information (MI) and conditional one (CMI). We then demonstrate the flexibility of MI and CMI for measuring compositional understanding, localizing words in images, and reflecting the change effects of prompt intervention in text-aware image editing. We conduct extensive experiments against competitive attention-baselines, and provide qualitative and quantitative results that demonstrate the advantages of INFO.

Reviewers qKVB, S4G8, and EMcD expressed confusion about the viability of our approach to image editing/controlled generation. Our experiments are designed to illustrate MI and CMI as powerful metrics to measure the fine-grained alignment between image and text. While this can be used to measure alignment with **standard** image editing techniques, as we show in Sec. 3.3, image editing is not the focus of our work.

Having stated our work’s objective, we would like to address your key concerns: clarifying our contributions, adding background information, improving presentation, and reproducibility.

# Contributions (qKVB, gEHf, U1vn)
## Scientific Findings
- Existing evaluation metrics significantly underestimate compositional understanding capabilities of diffusion models. Our INFO metric indicates that SD 2.1 WITHOUT fine-tuning outperforms previous work [1] on the ARO benchmark by a large margin, an observation that holds true even for SD 2.1 fine-tuned on composition-aware negatives! Our findings also support favorably the observation that CLIP checkpoints distilled from SD improve performances on the ARO benchmark. Accuracy (\%) of Stable Diffusion and its OpenCLIP backbone on the ARO Benchmark is in the following table (c.f. Table 1 in our manuscript):

 |      **Method**     | **VG-A** | **VG-R** | **COCO** | **FLickr30k** |
 |:-------------------|:--------:|:--------:|:--------:|:-------------:|
 |     OpenCLIP[3]     |   64.6   |   51.4   |   32.8   |      40.5     |
 |      DiffITM[2]     |   62.9   |   50.0   |   23.5   |      33.2     |
 | HardNeg-DiffITM*[2] |   67.6   |   52.3   |   34.4   |      48.6     |
 |     Info. (Ours)    | **72.0** | **69.1** | **40.1** |    **49.3**   |

*: conducts additional fine-tuning with compositional-aware hard negatives

- INFO is more capable of measuring change effects of image editing compared to attention-based methods. In Section 3.3, we conduct a prompt intervention experiment, and find change effects induced measured by INFO are more reflective of actual change size compared to attention-based methods.

- Neither INFO nor attention-based method is perfect for objective segmentation tasks, as illustrated in Section 3.2. However, we observe INFO can localize abstract concepts such as adjectives and adverbs more effectively.

[1] Krojer et al. 2023. Are Diffusion Models Vision-And-Language Reasoners? NeurIPS

[2] Franzese et al. 2023. MINDE: Mutual Information Neural Diffusion Estimation. ArXiv

[3] Ilharco et al. 2021. https://zenodo.org/records/5143773

## Methodological Contributions
Unlike widely adopted practices of using diffusion models (DMs) for text-to-image generation, we adapt DMs as powerful density estimators to quantify relationships between image and text, by leveraging the connection between the minimum mean-squared error (MMSE) and the MI/CMI discussed in Section 2.1. Our approach entails key distinctions from information-theoretic and attention-based counterparts:

- Compared to other estimators grounded in information theory [2], INFO is straightforward to implement and requires minimal changes to the existing APIs of Stable Diffusion (SD), while attention-based methods extract features from attention layers of UNet, probing deeply inside SD. We will release code as part of our project to contribute a novel estimator that reliably quantifies relationships between images and text.

- INFO computes principled estimates of images and texts, and better reflects the actual changes in the image after an intervention (c.f. Fig. 3 of our manuscript), however, attention does not (c.f. Fig. 1 of our manuscript).

- INFO is architecture-agnostic and can be adapted to measure relationships between any modality, as long as their synthesis is controlled by a DM that optimizes the MMSE. In this work, we study text-to-image diffusion models for their ease of visualization and plethora of baselines; but we hope to establish our INFO estimator as a strong candidate in other domains wherein DMs are valuable, such as robotic control and protein synthesis.

---

> ### Author Response · Authors · 2023-11-22
> **General Response (2/2)**
>
> # Mutual Information Estimator Preliminaries (EMcD, ajkd, S4G8)
> We appreciate the reviewers’ concern about the information theory background (or the lack thereof). We have updated our manuscript to include more preliminaries in Section 2.1 of our updated manuscript.
>
> # Presentation
> We thank the reviewers for their concerns about the presentation. We hope our contribution section in this general response, as well as the additional background on information theory in the updated manuscript, can alleviate some of your concerns. We will also update our paper to improve clarity.
>
> # Reproducibility
> We appreciate the reviewers’ interest in understanding our implementation and experiments in detail. To comply with anonymity, we did not include a link to our GitHub page, with open-source code that reproduces all experiments. However, we include our code in the supplementary material for the reviewers' benefit.

---

### Meta-Review · Area_Chair_EowG · 2023-12-06

**Metareview:**

- Claims and findings:
This work presents a new method for information decomposition at both pointwise and pixel-wise level via well-trained diffusion models, utilizing the results such that there is an exact connection between Mutual Information (MI) and MMSE denoising estimators under Gaussian noise, as well as the exact correspondence between denoising and density modeling. While the original MI is the average over the samples, the authors consider per-sample (point-wise) MI and further decompose it to per-pixel MI, which gives rise to a visualization of which region in the image is related with a certain word. In experiments, the proposed visualization method showed superior interpretation over the attention-based method.

- Strengths:

Reviewers highlight that the submissions introduces a novel approach to measuring the relationship between the text conditioning and the image pixels in text-to-image diffusion models. Authors show that their information-theoretic approach can be easily interpreted without additional overhead. In addition, experiments clearly demonstrates both the advantages and disadvantages of the proposed approach, providing a clear picture of when the mutual information can be useful and where attention can be used instead.


- Weaknesses:
Reviewers have pointed out that the submission studies interpretability for text-to-image diffusion models but there is no clear demonstration of where this could be applied other than better visualizations of text-image relations. In addition, reviewers have also noted that the walkthrough of some key results that help construct the proposed information decomposition approach is too brief.

- Missing from submission:
As reviewer qKVB points out it would be great to have a study of what these interpretability results can unlock in terms of better aligning text-to-image models.

**Justification For Why Not Higher Score:**

There are some weaknesses related to practicality of the interpretability technique. In addition, some of the key results are briefly discussed.

**Justification For Why Not Lower Score:**

Reviewers collectively agree that this paper is slightly over the acceptance threshold.

---

### Decision · Program_Chairs · 2024-01-16

Accept (poster)